

# Navigating the complexities of the forest land sharing *vs* sparing logging dilemma: analytical insights through the governance theory of social-ecological systems dynamics

Jean-Baptiste Pichancourt

Université Clermont-Auvergne (UCA), INRAE, Laboratoire d'Ingénierie des Systèmes Complexes (UR LISC), Institut National de la Recherche pour l'Agriculture, l'Alimentation et l'Environnement (INRAE), Clermont-Ferrand, Aubière, France

Corresponding author
Jean-Baptiste Pichancourt,
jean-baptiste.pichancourt@inrae.fr

## ABSTRACT

This study addresses the ongoing debate on forest land-sparing *vs* land-sharing, aiming to identify effective strategies for both species conservation and timber exploitation. Previous studies, guided by control theory, compared sharing and sparing by optimizing logging intensity along a presumed trade-off between timber yield and ecological outcomes. However, the realism of this trade-off assumption is questioned by ecological and governance theories. This article introduces a mathematical model of Social-Ecological System (SES) dynamics, distinguishing selective logging intensification between sharing and sparing, with associated governance requirements. The model assumes consistent rules for logging, replanting, conservation support, access regulation, socio-economic, soil and climate conditions. Actors, each specialized in sustainable logging and replanting of a single species, coexist with various tree species in the same space for land sharing, contrasting with separate actions on monospecific stands for sparing. In sharing scenarios, a gradient of intensification is created from 256 combinations of selective logging for a forest with eight coexisting tree species. This is compared with eight scenarios of monospecific stands adjacent to a spared eight-species forest area safeguarded from logging. Numerical projections over 100 years rank sparing and sharing options based on forest-level tree biodiversity, carbon storage, and timber yield. The findings underscore the context-specific nature of the problem but identify simple heuristics to optimize both sparing and sharing practices. Prioritizing the most productive tree species is effective when selecting sparing, especially when timber yield and biodiversity are benchmarks. Conversely, sharing consistently outperforms sparing when carbon storage and biodiversity are main criteria. Sharing excels across scenarios considering all three criteria, provided a greater diversity of actors access and coexist in the shared space under collective rules ensuring independence and sustainable logging and replanting. The present model addresses some limitations in existing sparing-sharing theory by aligning with established ecological theories exploring the intricate relationship between disturbance practices, productivity and ecological outcomes. The findings also support a governance hypothesis from the 2009 Nobel Prize in Economics (E. Ostrom) regarding the

positive impact on biodiversity and productivity of increasing polycentricity, *i.e.*, expanding the number of independent species controllers' channels (loggers/replanters/supporters/regulators). This hypothesis, rooted in Ashby's law of requisite variety from control theory, suggests that resolving the sharing/sparing dilemma may depend on our ability to predict the yield-ecology performances of sparing (in heterogeneous landscapes) *vs* of sharing (in the same space) from their respective levels of "polycentric requisite variety".

## INTRODUCTION

The land sparing-sharing debate provides insights into how to organize the coexistence between the largest possible diversity of species that need to be conserved, and the largest possible diversity of resource appropriators who need some or all of these species for a variety of goods and services. This debate unfolds across various land types and coastal waters, presenting two contrasting perspectives on the interaction between humans and nature within social-ecological systems. Each viewpoint comes with its distinct advantages and drawbacks, as outlined by *Phalan et al. (2011)*, *Fischer et al. (2014)*, *Dressler et al. (2016)* and *Phalan (2018)*.

In forestry systems, land-sparing logging emphasizes spatial segregation at the landscape scale for various purposes such as production, conservation of biological diversity, and recreation. The ultimate goal of sparing is to efficiently manage distinct, mono-specific stands, each dedicated to specific livelihood benefits. Examples can include different mono-specific stands of local timber species (*Lamb, 1998*), parcels of fast-growing mono-specific carbon planting of mangrove or pinus (*Alongi, 2012*; *Pariyar et al., 2019*), parcels of deciduous tree species for groundwater re/dis-charge and purification (*Schwaiger et al., 2018*), and parcels of local species for food, firewood, medicine, or other non-timber forest products (*Wong, 2000*), among others. By adopting a land-sparing logging approach, it is expected that actors can collectively agree to save the largest possible and most contiguous forest area for maximizing the conservation of biodiversity and other intangible ecosystem services outcomes. In contrast, land-sharing logging advocates for a locally integrated approach to these social and ecological diversities on the same forest land, in order to selectively and sustainably log the right set and quantity of tree species (*Butsic et al., 2012*; *Runting et al., 2019*). This approach involves playing with the disturbance intensity, frequency and functional complementarities between wild and domestic tree species (*Pichancourt, 2023*). Intermediate land-use strategies can also be implemented that consist in mixing the two land-use logging practices at the landscape scale (*e.g.*, *Mastrangelo & Laterra, 2015*; *Runting et al., 2019*).

Methodologies have been devised to determine the optimal land-use strategy, whether it involves sharing, sparing, or falls within an intermediate approach (*Phalan, 2018*). These methodologies generally hinge on a core principle, which involves characterizing the nature of the relationship between yield and environmental outcomes (such as levels of timber extraction *vs* biodiversity). Optimization techniques, based on control theory, are then employed to identify the most effective analytical solution (*Green et al., 2005*; *Butsic et al., 2012*; *Fischer et al., 2014*). In situations characterized by a biodiversity-yield trade-off, *i.e.*, where the maximization of one dimension occurs when the other is minimized, the preference for land-sparing emerges. Conversely, if the maximum occurs at intermediate solutions, the preference is for land-sharing. However, relying solely on this approach and empirical data reveals, through successive meta-analyses, that there is no one-size-fits-all solution (*Phalan et al., 2011*; *Kremen, 2015*; *Luskin et al., 2018*; *Phalan, 2018*). These meta-analyses show that performance of sharing, sparing, and intermediate solutions is indeed influenced by various factors, including land-use objectives (*e.g.*, maximizing different types and the number of ecosystem services or the diversity of different taxa), constraints (*e.g.*, spatial scale, institutional arrangements), and modes of action (*e.g.*, land-use intensity). Recent empirical and modeling studies have attempted to combine and assess the relative impact of these different factors on the yield-conservation relationship. These efforts aim to infer whether exact analytical solutions or rules of thumb could guide the selection of the most effective land strategy (*Phalan, 2018*).

For example, altering the model of land-use intensity (*e.g.*, linear *vs* non-linear, lower disturbance *vs* intermediate disturbance hypothesis) is expected to modify the shape of the yield-biodiversity trade-off, influencing the optimal outcome in the sparing/sharing/intermediate continuum (*Butsic et al., 2012*). Applying this concept to empirical data in an Indonesian forest logging system, *Runting et al. (2019)* demonstrated that the optimal land strategy should not only vary with taxa but also with the intensity of selective logging practices. This leads to recommendations for minimal intensity logging toward the sparing end of the continuum. On the contrary, in a large Catalonian forested landscape (*Marull et al., 2015*, *2018*), found that coupling intermediate land-use intensity with an intermediate sharing-sparing complexity level should result in maximum diversity in bird taxa. Meanwhile, mammal biodiversity is maximized in fully spared, low-intensively disturbed landscapes, and vascular plant diversity reaches its maximum when shared forests (*i.e.*, under moderately intensive exploitation) dominate the landscape.

Inspired by the recommendations of *Fox (2013)*, *Barabás, D'Andrea & Stump (2018)*, other studies delved deeper into the effects of land-use intensity processes–like selective logging—on the relationship between tree species coexistence, timber yield and carbon sequestration in land sharing. For instance, *Pichancourt et al. (2014)*, *Pichancourt (2023)* decomposed the land-use intensification strategy into four selective logging disturbance practices: (i) greater logging frequency on the most dominant species, (ii) greater logging intensity on the most dominant species, (iii) sustainable selective logging on all dominant tree species, and (iv) smaller to greater diversity of selective logging targets. Within the model, a greater diversity of selective logging practices became apparent when allowing a

higher number of independent loggers access to the forest commons. In the model, each logger was authorized to act for its own interest under strict governance rules, by selectively yet sustainably log and replant a single tree species among those coexisting. This process of intensification differed significantly from the unselective logging strategy assumed in previous models. With the model it was demonstrated that a greater number of loggers could be compatible with high levels of biodiversity and productivity. In their theoretical pursuit to comprehend the joint effect of governance and ecological laws on the efficacy of land-sharing system, *Pichancourt et al. (2014)* and *Pichancourt (2023)* introduced skepticism regarding the universal applicability of the relationship between land-use intensification and the assumed trade-off among tree diversity, timber yield, and above-ground carbon storage. Naturally, this prompts inquiries into the implications for the sparing-sharing debate and the foundational assumptions guiding it. However, given the importance of the governance structure and parameters on logging intensification when constructing their model (*Pichancourt et al., 2014*; *Pichancourt, 2023*), it follows that they may also play an important role when comparing the outcomes of SES models of land-sparing and land-sharing.

The existing body on governance theory, exemplified by works such as those by *Ostrom (1990)*, *Agrawal, Wollenberg & Persha (2014)*, *Jiren et al. (2018)*, *Crespin & Simonetti (2019)*, offers valuable insights into comprehending the distinctions of governing logging intensification between land-sharing and land-sparing approaches. On one hand, land-sparing rules and collective-choice arrangements ought to address spatial conflicts arising in multifunctional heterogeneous landscapes. These conflicts typically involve public conservation agencies or communities managing protected areas and independent private or community loggers who own and oversee adjacent monocultural lands. On the other hand, regulations and collective-choice frameworks for land-sharing are crafted to manage conflicts emerging among a diversity of competing loggers on the same forest land over a diversity of species viewed as common-pool resources (*Natcher, Davis & Hickey, 2005*; *Lemos & Agrawal, 2006*). This encompasses the oversight of access to common-pool tree resources, logging, replanting, and other supporting actions of species, as well as monitoring and penalizing free riders in the commons.

While these qualitative insights provide general information, they do not offer methodological guidelines for effectively implementing these distinctions when constructing quantitative SES models to compare land-sharing and land-sparing scenarios. A possibility is to use the Coupled Infrastructure System framework (CISF) as a guiding tool (*Anderies, Janssen & Ostrom, 2004*; *Anderies, 2015*; *Anderies, Barreteau & Brady, 2019*; *Muneepeerakul & Anderies, 2020*) to analytically frame this problem and compare it with past analytical models of sharing-sparing. Historically, the CISF, developed by Elinor Ostrom (2009 Nobel Prize Laureate in Economics), served to depict the systemic nature of governing common pool resources in social-ecological systems (SES) (*Anderies, Janssen & Ostrom, 2004*). Its structure typically involves up to four interacting social and ecological compartments, representing infrastructures and/or actors. This simplicity allows the

transformation of any SES into a set of mathematical equations, facilitating its examination as a complex system (*Anderies, 2015*; *Muneepeerakul & Anderies, 2020*). Moreover, it enables predictions regarding dynamics and robustness under various scenarios of external and internal changes (*Muneepeerakul & Anderies, 2017*; *Homayounfar et al., 2018*; *Muneepeerakul & Anderies, 2020*; *Houballah, Mathias & Cordonnier, 2021*; *Houballah, Cordonnier & Mathias, 2023*). The CISF approach, tightly linked to Ostrom's governance theory of common pool resources (CPR), the Institutional Analysis and Development Framework (IADF), and the social-ecological system framework (SESF), is now widely recognized as a pivotal method for analytically framing the mechanistic and causal problems of CPR governance in SES (*Anderies, Barreteau & Brady, 2019*; *Bernstein et al., 2019*; *Cumming et al., 2020*). However, it has never been used to frame the analytical problem of sharing *vs* sparing.

In a prior publication (*Pichancourt, 2023*), I delved into the intricacies of forest land-sharing, shedding light on the fundamental social-ecological constraints influencing the coexistence between diverse tree species and independent selective loggers. Notably, that study did not explore findings related to land-sparing. The current article adopts a more pragmatic stance, introducing a method specifically crafted to enable a comparative analysis of the efficacy of both land-sharing and land-sparing systems. This is accomplished by employing an extended version of the analytical theory of CISF.

Building on this robust analytical foundation, the article investigates the potential of the CISF to effectively delineate the fundamental distinction between land-use intensification processes in both land sharing and land sparing, and their consequences on the sparing-sharing ranking based on forest yield and ecological outcomes. To achieve this, the article initially illustrates the CISF differences between land-sharing and land-sparing. It then outlines the transformation of these differences into distinct sets of equations, capturing their dynamics and showing where assumptions are made in relation with previous literature on sharing and sparing. Furthermore, the article showcases the integration of more refined models from *Pichancourt et al. (2014)* and *Pichancourt (2023)* to address the complexities associated with logging intensification and its control in the presence of diverse actors engaged in various activities with different tree species– specifically, selective logging, replanting, access regulation, and species support and monitoring. The article leverages these two separate systems of equations to conduct numerical analyses and rank various scenarios of land sparing and land sharing, based on their tree biodiversity, above-ground carbon and timber yield outcomes across. Subsequently, the article derived streamlined heuristics for assessing the performance of one land-use system vis-à-vis the other. These results were then examined in the broader agenda of developing such models to solve social-ecologically richer trade-off problems associated with the sharing-sparing nexus, as outlined by *Fischer et al. (2014)*, *Mastrangelo & Laterra (2015)*, *Phalan (2018)*. Furthermore, this discussion explored potential pathways toward uncovering a more overarching theory capable of elucidating and endorsing the choice between these two types of land-use strategies.

## METHOD

### General structural differences between the SES model of land-sharing *vs* sparing

The underlying framework of the forest SES dynamics model adheres as much as possible to the coupled infrastructure system framework (CISF) guidelines, as initially outlined by *Anderies, Janssen & Ostrom (2004)* and further refined in subsequent works (*Anderies, 2015*; *Anderies, Barreteau & Brady, 2019*). However, certain adjustments were made to address limitations identified by *Bernstein et al. (2019)* and *Pichancourt (2023)*. Notably, the adaptation involved a simplification of the CIS model, focusing on three interconnected compartments while omitting the one dedicated to policy-making actors, often denoted as public infrastructure providers (PIP) in the literature. Consequently, some linking processes–typically encoded as 2a, 2b, 3a, and 3b in the CISF–were not explicitly depicted in this model, as illustrated in Fig. 1. The three compartments in this context are:

- The forest resource species and infrastructure (RSI) delineate the physical factors, ecological composition, demographic structure, and dynamics of a coexisting group of species and ecosystem processes, encompassing carbon cycles, soil water dynamics, and deadwood dynamics.
- The exploiting actors and infrastructures (EAI, often referred to as resource users in CISF studies) have the primary objective of seeking and maintaining direct resource benefits from the RSI. Here, resource exploiters are involved in activities such as timber harvesting, post-harvest management, and species monitoring, driven solely by individual profit motives. This is done without consideration for the impacts on other co-vulnerable loggers and the state of forest variables crucial for public or common-good ecosystem services, including biodiversity levels and carbon storage for climate mitigation.
- The supporting actors and infrastructures (referred to as SAI here, and commonly known as public infrastructures in CISF studies) aim to sustain the condition of the RSI and promote public and common-good ecosystem services, such as carbon storage and biodiversity. Consequently, they undertake actions to support the RSI and EAI state, regulation of access the the RSI, and monitoring the state of the RSI, of EAI and of ecosystem services.

It is evident that both RSI and EAI compartments are conventionally employed in analytical studies to assess the yield/conservation trade-off discussed in the sharing-sparing debate (*e.g.*, see *Green et al., 2005*; *Butsic et al., 2012*; *Marull et al., 2015*, *2018*; *Runting et al., 2019*). However, it should be noted that the SAI is not used for this purpose.

Now using this CISF approach with the three compartments, and following the work initiated in (*Pichancourt, 2023*), we can tentatively represent the governance differences between forest land-sharing and sparing (Fig. 1). Then using standardized guidelines

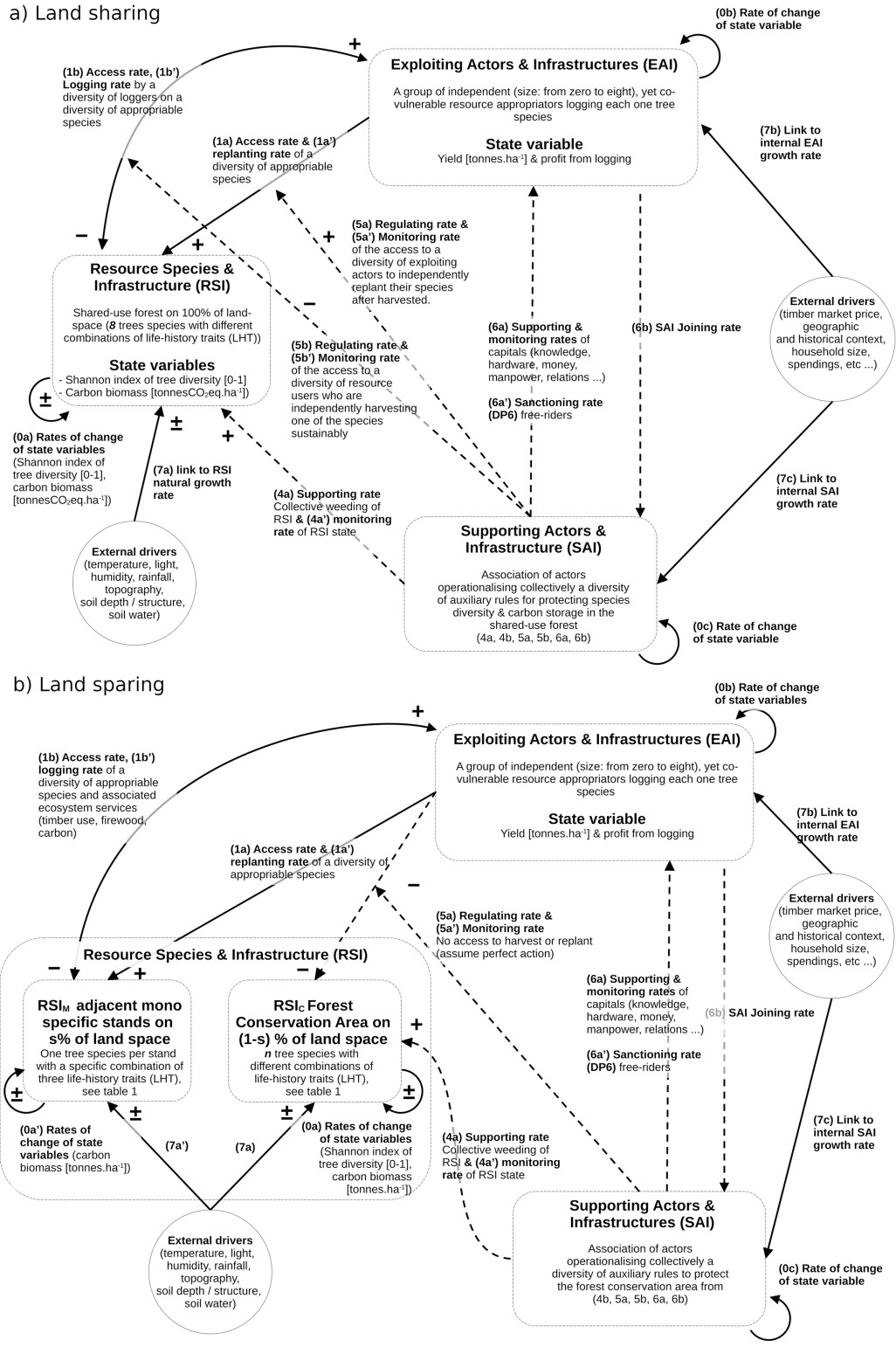

**Figure 1** Qualitative SES models for land-sharing (A) and land sparing (B), in line with the encoding used for the coupled infrastructure system framework (CISF). The three interacting compartments (one ecological RIS and two social EAI and SAI) are defined by state variables and interactive controls (1A–6B). The models are then transformed into sets of equations using standardized CISF methodology developed in the method section.

described by *Anderies (2015)*, *Muneepeerakul & Anderies (2020)* and the nomenclature of variables described in Fig. 1, these two CISF representations could be transformed into distinct systems of equations.

### General model of SES dynamics for land-sharing

On the one hand, the CISF land-sharing model (Fig. 1A) could be mathematically described as follows:

$$\frac{dRSI}{dt} = \pm \overbrace{7a(RSI(t)).RSI}^{\text{Natural Growth}} + \overbrace{4a(RSI(t)).4a'(RSI(t)).RSI.SAI}^{\text{Support}}$$
$$- \overbrace{1b(RSI(t)).1b'(RSI(t)).EAI.RSI}^{\text{Access \& Logging}} . \overbrace{5b(t).5b'(t).SAI}^{\text{Regulated}} \qquad (1a)$$
$$+ \overbrace{1a(RSI(t)).1a'(RSI(t)).EAI.RSI}^{\text{Access \& Replanting}} . \overbrace{5a(t).5a'(t).SAI}^{\text{Regulated}}$$

$$\frac{dEAI}{dt} = - \overbrace{7b(t).EAI}^{\text{Internal Loss}} + \overbrace{1b(RSI(t)).RSI}^{\text{Access \& Logging}} . \overbrace{5b(t).5b'(t).SAI}^{\text{Regulated}} + \overbrace{(6a(t) - 6a'(t)).EAI.SAI}^{\text{Support \& Sanction}} \quad (1b)$$

$$\frac{dSAI}{dt} = - \overbrace{7c(t).SAI}^{\text{Internal Loss}} + \overbrace{6b(t).EAI.SAI}^{\text{Joining}} \qquad (1c)$$

   In this system of equations, the state of RSI and its dynamics $\left(\frac{dRSI}{dt} = 0a\right)$ can be conceptualized as representing the aboveground (carbon) biomass $[tonnes(CO_2eq).ha^{-1}]$ and the Shannon index of tree diversity $[0-1]$. These states are primarily influenced by the access rates to a diversity of loggers for replanting (1a) and logging (1b), by the selective logging intensity (1b'), and the post-harvesting replanting rates (1a'). The state of EAI represents the value of timber yield or economic capital $[.ha^{-1}$ or $tonnes.ha^{-1}]$ obtained through selective logging (1b') by the diverse group of loggers accessing through 1b. Loggers can invest a portion of their EAI economic capital at a certain rate into the SAI (6b) to implement collective-choice actions. These SAI actions involve collective monitoring and support of the state of RSI (respectively 4a, 4a'), regulation, and monitoring rates of its access 1a and 1b (*via* 5a, 5b, 5a',5b'), and the rate of collective support they can receive under conditions to be met to avoid sanctions (6a, 6b). Given the information presented in the introduction, the $\pm$ signs in the set of equations denote the lack of prior knowledge regarding the directional impact of the disturbance regimes imposed on the tree species–such as through logging (1b'), replanting (1a'), supporting actions (4a), or environmental variables (7a)–may have directly on the natural growth rate of the RSI in response to environmental variables (7a), *i.e.*, on tree biodiversity and above-ground organic (carbon) biomass. As it was introduced, due to complex inter-species relationships, it is not because we extract individuals of a given species that the forest-level biomass and species diversity growth rates will be negatively affected. Conversely, it is not because we perfo\rm supporting \, actions by replanting or weeding a given tree species that the same forest state growth rates will be positively affected. Not making any a-priori regarding the sign and value is one of the major difference with

previous analytical studies as outlined in the introduction (see *e.g.*, *Green et al., 2005*; *Butsic et al., 2012*; *Marull et al., 2015*, *2018*; *Runting et al., 2019*).

***General model of SES dynamics for land-sparing***

By comparison in the context of land-sparing (Fig. 1B), and without assuming in the general case pefect protection of the conservation zone ($RSI_C$), the CISF model leads to another system of equations where we now need to distinguish in the RSI, the dynamics of $RSI_C$ and of $RSI_M$. In this case, the exploiting actions are only applied to $RSI_M$, whereas the collective supporting and regulating actions (4a, 4b, 5a, 5b) are only applied to $RSI_C$. By making the same asumptions regarding the impact of disturbances on the natural growth of both the monospecific and the spared conservation area, we obtain the following set of equations for land-sparing:

$$\frac{dRSI_C}{dt} = \pm \overbrace{7a(RSI_C(t)).RSI_C}^{\text{Natural Growth}} + \overbrace{4a(RSI(t)).4a'(RSI_C(t)).RSI_C.SAI}^{\text{Support}} + \overbrace{7a''.RSI_M}^{\text{Seed Colonization}}$$
$$- \overbrace{1c(RSI_C(t)).1c'(RSI_C(t)).EAI.RSI_C.5b(t).5b'(t).SAI}^{\text{IllegalAccess \& Logging} \qquad \text{Regulated}} \tag{2a}$$

$$\frac{dRSI_M}{dt} = \pm \overbrace{7a'(RSI_M(t)).RSI_M}^{\text{Natural Growth}} + \overbrace{7a'''.RSI_C}^{\text{Seed Colonization}}$$
$$- \overbrace{1b'(RSI_M(t)).EAI.RSI_M}^{\text{Logging}} + \overbrace{1a'(RSI_M(t)).EAI.RSI_M}^{\text{Replanting}} \tag{2b}$$

$$\frac{dEAI}{dt} = - \overbrace{7b(t).EAI}^{\text{Internal Loss}} + \overbrace{1b'(RSI_M(t)).EAI.RSI_M}^{\text{Logging}} + \overbrace{(6a(t) - 6a'(t)).EAI.SAI}^{\text{Support \& Sanction}} \tag{2c}$$

$$\frac{dSAI}{dt} = - \overbrace{7c(t).SAI}^{\text{Internal Loss}} + \overbrace{6b(t).EAI.SAI}^{\text{Joining}} \tag{2d}$$

## Model simplifications to match the analytical framework used in the sharing-sparing debate

Considering the rich processes outlined in the systems of equation, attempting to examine the impacts of all of them within a single article would be impractical. This article focuses on the investigation of select processes to facilitate comparison with previous analytical studies. Others that deviate more from previous studies and our original question are addressed in the discussion section, by delving into the interest to strategically constrain and relax certain assumptions, aiming to enhance our understanding of specific aspects pertaining to the land-sharing *vs* land-sparing debate. Below are the simplifying asumptions that were considered.

## Simplified model of SES dynamics for land-sharing to match literature

Firstly, even if the three compartments (RSI, EAI, SAI) are important to understand the interest of governance in the sparing-sharing debate, their feedback loops aren't, and were also not included in previous analytical work when considering RSI and EAI. For instance in previous studies (see *Green et al., 2005*; *Butsic et al., 2012*; *Marull et al., 2015*, *2018*; *Runting et al., 2019*), the state of EAI does not exert feedback on the state and dynamics of RSI and SAI, given the absence of available information on the extent to which the value of timber yield influenced reinvestment in individual (1a', 1b') and collective actions (4a, 4a'). Consequently, the SAI initial conditions were considered time-invariant $\left(\frac{dSAI}{dt} = 0 c = 0\right)$ and consistent across all scenarios for both sharing and sparing.

The access control rate 5b (link 5b in Fig. 1), controlled by SAI, and the access monitoring rates for access, logging and replanting were also assumed to be perfect in these studies (5a = 5b = 5a' = 5b' = 1). Simultaneously, perfect monitoring (linking variable 4a = 1) of the state of biodiversity and carbon storage in the conservation area was assumed. Furthermore regulation of the access for logging and replanting are considered the same (5a = 5b), such that loggers replant straightaway after logging. Finally, seed colonization between the monocultural stand and the spared conservation area were not considered in previous studies (*i.e.*, 7a" = 7a''' = 0) and thus not in this one. Equation (1) could thus be simplified to match more existing analytical studies, such that:

$$
\frac{dRSI}{dt} = (\pm \overbrace{7a(RSI(t))}^{\text{Natural Growth}} + \overbrace{4a(RSI(t)).4a'(RSI(t))}^{\text{Support}}
$$
$$
- \underbrace{1b(RSI(t)).1b'(RSI(t))}_{\text{Access \& Logging}} . \overbrace{5b(t)}^{\text{Regulated}} + \underbrace{1a(RSI(t)).1a'(RSI(t))}_{\text{Access \& Replanting}} . \overbrace{5b(t)}^{\text{Regulated}} ).RSI \tag{3a}
$$

$$
\frac{dEAI}{dt} = \underbrace{1b(RSI(t)).1b'(RSI(t)).RSI.EAI}_{\text{Access \& Logging}} . \overbrace{5b(t)}^{\text{Regulated}} \tag{3b}
$$

## Simplified model of SES dynamics for land-sparing to match literature

For the land-sparing previous studies also assumed that there was no biological flow between the conservation zone and the monospecific stand used for either timber plantation or carbon plantation (*i.e.*, 0a = 0a' = 0 in Fig. 1). They also assumed perfect monitoring and control of the access into the conservation area (5a = 5b = 1 and 1c = 0). This perfect monitoring implied that loggers from EAI could not access and freeride by illegaly logging the preserved forest conservation zone in spared $RSI_C$ (1b = 1b' = 1a = 1a' = 0). Regarding monocultural plantations $RSI_M$, all monitoring, support, and regulatory actions were presumed to be individually conducted by the appropriator from EAI through linking variables 1a' (logging) and 1b' (post-harvesting replanting), and by also assuming perfect monitoring. Equation (2) could thus be simplified to match more existing analytical studies, such that:

$$\frac{dRSI_C}{dt} = (\pm \overbrace{7a(RSI_C(t))}^{\text{Natural Growth}} + \overbrace{4a(RSI_C(t)).4a'(RSI_C(t))}^{\text{Support}}).RSI_C \quad \text{(4a)}$$

$$\frac{dRSI_M}{dt} = (\pm \overbrace{7a'(RSI_M(t))}^{\text{Natural Growth}} - \overbrace{1b'(RSI_M(t))}^{\text{Logging}} + \overbrace{1a'(RSI_M(t))}^{\text{Replanting}}).RSI_M \quad \text{(4b)}$$

$$\frac{dEAI}{dt} = \overbrace{1b'(RSI(t)).RSI_M.EAI}^{\text{Logging}} \quad \text{(4c)}$$

## Emerging analytical limits and suggested path forward

Now that we have developed simplified systems of equations that more closely align with previous analytical studies, we could in theory seek to compare them with those from earlier research, particularly the work of *Butsic et al. (2012)*, which can arguably be regarded as the most comprehensive analytical study useful for the sharing-sparing debate.

However, the introduction of $\pm$ signs complicates the deduction of equilibrium and stability solutions for both sharing and sparing, both at a general level and locally. Consequently, adhering to the guidance provided by *Pichancourt et al. (2014)* and *Pichancourt (2023)*, we opted for an alternative approach. This involved disaggregating the systems of equations to create a more detailed microscopic model that delineates the intricate relationships between species, actors, and their actions. The aim is to subsequently unveil the "true" relationship (replacing the $\pm$ sign) between disturbance actions, timber yield, tree biodiversity, and above-ground living (carbon) biomass for both sparing and sharing.

Following this strategy, a concise summary of the mathematical decomposition for each compartment and its associated set of actions is presented for both land-sharing and land-sparing. The mathematical framework of the RSI model was originally formulated by *Pichancourt et al. (2014)*, while those associated with the EAI and SAI were developed in *Pichancourt (2023)*.

## Micro-model of resource species & infrastructure (RSI)

The state variable of the shared *RSI* and spared $RSI_C$ presented in Eqs. (3a), (4a) and (4b) were defined using a deterministic mathematical model of a multi-species forest ecosystem dynamics described in the SI material link provided in the original reference (*Pichancourt et al., 2014*), accessible *via* the original url provided at the end of this article.

This model estimates the demographic structure and dynamics of trees (density, survival, growth, seed production, and germination rates) defined by their life stage (seeds, seedlings, juvenile and adult trees), their size, and different combinations of species functional traits (specific leaf area, specific timber density, seed size), and constrained by specific environmental parameters related to forest soil (type, depth, topography), and climate-related variables affecting tree physiology (*e.g.*, temperature, rainfall, humidity, vapor pressure, atmospheric irradiance scenarios). From this model, we can estimate the

annual or asymptotic population growth rate and demographic structure of a single tree species. It can also deduce the demographic outcomes of coexisting species, encompassing both intra-and inter-species competitive interactions, and driven by limiting conditions such as soil water competition and vertical competition for light. Through these ecological processes, the model can then estimate annual forest state variables, *e.g.*, above-ground organic carbon biomass (tonnes $CO_2$ eq/ha) and the Shannon index of species diversity. These can then be used here to represent the RSI state variable in Eqs. (3a), (4a) and (4b).

Similar to *Pichancourt (2023)*, the RSI micro-model was initiated with the same ecological and environmental parameters. Firstly, forests were composed of eight coexisting species for computational reasons. Details regarding the eight species employed in the model with respective caracteristic combinations of the species functional traits can be found in Table 1. It also adhered to the same parameterization outlined in *Pichancourt et al. (2014)*, along with consistent initial density profiles for various species and tree size classes (*Pichancourt, 2023*), that reflect what is practiced for aforestation projects. In practice, the simulations were initiated with 5 $saplings.m^{-2}$ planted (all species combined), given a practice usually from 1 to 30–50 $saplings.m^{-2}$ for respectively low and high density planting. The model also reduced the number of environmental constraints by considering the same climate and landscape variables as in *Pichancourt (2023)*, specifically wetter conditions without soil water limitations over a 100-year simulation. By doing so, the model was expected to produce more salient results between different scenarios of monospecific stands and land-sharing (*Pichancourt, 2023*). Finally, the environmental parameters were similarly set to correspond to realistic conditions commonly found in sub-tropical areas for soil quality and depth (3 m deep, 50% sand, 30% clay, 20% silt), climate from Queensland, Australia (−27.467778, 153.027778), and no water limitation affecting the growth and competition of trees (*e.g.*, a lowland forest).

In the context of land-sharing (Fig. 1A), the RSI model was employed to project, on an annual basis, the dynamics of the eight coexisting species spanning a duration of 100 years. Each projection encompassed scenarios with and without distinct logging and controls, as elucidated in the EAI and SAI micro-models detailed in the subsequent sub-sections. In the context of land-sparing (Fig. 1B), projections occurred in two phases. Initially, the dynamics of the eight coexisting species were projected for $RSI_C$, wherein they were shielded from any logging and planting (link 1a controlled by 5a, 5b detailed in Fig. 1B). Subsequently, the dynamics were projected for $RSI_M$ for each of the eight monospecific stands, as outlined in Table 1. This approach allowed for the comparison of the statistics of the eight monospecific stands, enabling an assessment of the eight possible land-sparing scenarios and the other land-sharing scenarios described in the next subsection.

## Micro-model for the exploiting actors & infrastructures (EAI): log, replant

The EAI micro-model delineated a framework of conditions and equations governing the impact of diverse resource appropriators accessing the RSI (1a, 1b) independently to monitor and/or sustainably log their tree species of interest (1b'). Subsequently, they would replant the logged species (1a'). Following logging, the model computed timber profit for
**Table 1 Combinations of independent life-history traits that characterize eight archetypal species, defined by their values of specific timber density (STD), specific leaf area (SLA) and seed size (SSS).** From the eight combinations of trait values, the model produced eight species with different morphologies, physiological traits, vital rates across their life-cycles, and population growth rates, as per *Pichancourt et al. (2014)*, *Pichancourt (2023)*. Simulations were run over 100 years to predict carbon storage and timber yield using conditions described in the method section (soil, water, climate, planting, harvesting).

| Species | Plant functional traits | | | Average mono-specific-stand level performances | |
| --- | --- | --- | --- | --- | --- |
| | STD $[kg.m^{-3}]$ | SLA $[m^2.kg^{-1}]$ | SSS [kg] | Carbon storage $[tC.ha^{-1}]$ | Annual timber yield $[t.ha^{-1}.y^{-1}]$ |
| sp#1 | 400 | 20 | $10^{-7}$ | 34.2 | 2.79 |
| sp#2 | 400 | 20 | $10^{-4}$ | 15.7 | 0.48 |
| sp#3 | 400 | 2.85 | $10^{-7}$ | 149 | 6.25 |
| sp#4 | 400 | 2.85 | $10^{-4}$ | 30.7 | 1.11 |
| sp#5 | 1,000 | 20 | $10^{-7}$ | 51.7 | 0.02 |
| sp#6 | 1,000 | 20 | $10^{-4}$ | 16.4 | 0.07 |
| sp#7 | 1,000 | 2.85 | $10^{-7}$ | 111 | 0.03 |
| sp#8 | 1,000 | 2.85 | $10^{-4}$ | 67 | 0.28 |

each logger, taking into account the international timber market price, specific timber density (STD), and a forestry economic law detailed in *Pichancourt (2023)*. This study did not integrate social criteria related to timber harvest or biodiversity. Nonetheless, the methodology maintained consistency regardless of the quantity and nature of variables under comparison.

Within the model, each logger accessing the forest through 1a or 1b specialized in only one species, forming a distinct dyad. In this dyad, the logger focused exclusively on logging (1b') and replanting (1a') their chosen species. Logging and replanting rates for each dyad were modeled using econometric equations reflecting real-life decision-making processes, from a blend of empirical studies culled from the literature in different countries. Logging rates per tree in 1b' were determined using the equation associated with the model of timber harvesting from the method section of *Pichancourt (2023)*, accessible *via* the original url provided at the end of this article (section: "Model of timber harvesting"). It is crucial to emphasize that these equations accurately capture genuine logging behaviors, aligning the logging rate with the biology and biomass state of each species to guarantee its sustainability. Potential spillover effects resulting from logging on other tree species and competitive mechanisms are then accounted by the forest model described in the previous section. For this reason, we can regard loggers as genuinely independent in their practices, yet interconnected through species interactions and susceptible to shared vulnerabilities in their outcomes. Similarly, post-harvest planting in 1a' was defined using the equation presented in the method section from the same reference and was assumed to be independent between species (section: "Behavioral model for individual post-harvesting planting"), so the same independence and co-vulnerability applies here. The parameter values for these equations were also fixed and identical to those described in the reference.

These processes assumed a dependence to key constraining state variables within the RSI compartment, encompassing forest biomass, species life-history traits, and tree size

class (refer to Table 1). Additionally, they were influenced by external contextual socio-economic factors (link 7b in Fig. 1), including timber market price, household size, governance system type (private, public, community), and other geographic and historical considerations.

In the context of land-sharing (see Fig. 1A), the potential involvement of anywhere from zero to eight independent loggers coexisting in the same land introduced a myriad of possibilities encompassing various species combinations. This resulted in a total of $2^8 = 256$ potential dyadic combinations associated with logging activities. These combinations were used to estimate the global gradient of values for the logging rate. But compared to previous studies on sharing-spaing, factors such as the number of loggers, the variety of species logged, and variations in logging intensity per species collectively contributed to this richer logging intensification gradient. To gauge their respective impacts on RSI dynamics, independent simulations were conducted.

In the context of land-sparing (refer to Fig. 1B), distinct simulations were carried out for each of the eight monospecific stands. Utilizing identical equations for harvesting and replanting, akin to those in scenarios 1a and 1b, the values were estimated to understand their implications.

## Micro-model for the supporting actors & infrastructures (SAI): support, regulate, monitor

The SAI micro-model consisted also of a set of empirically derived econometric equations that predicted the likelihood of actors engaging in SAI activities, along with their allocated units of effort to support RSI state variables. The process of collective thinning/weeding in 4a' was also influenced by the socio-economic and physical context, whose relation was determined using the equations and associated parameters detailed in *Pichancourt (2023)* and accessible *via* the original url provided at the end of this article (section: "Collective weeding and thinning model"). Variables 5a, 5b, 5a', and 5b' were not defined this way, but rather encapsulated the 256 initial conditions of simulations governing the previously described intensity gradient.

## Method for comparing the performance between land sharing and sparing

The model was used to project the RSI and EIA state variables per hectare over a span of 100 years for each monospecific stand. Subsequently, the same 100-year projections were generated for all 256 possible combinations of land-sharing systems. Even if the model was aspatial, it was still possible to compare these SES performances between sharing and sparing using the following general principle of analysis. To better understand how we could make this comparison, let's focus on timber production as the first key criterion of comparison, by establishing the requirement that any monospecific land area (expressed as percentage of land occupied) should yield at least as much timber as any geographic unit 100% occupied by a land-sharing system. With this premise, consider a monospecific stand comprised of species sp#4 (refer to Table 1) being projected to produce twice the amount of timber, compared to all potential land-sharing systems where three resource
appropriators are sustainably and independently logging any combination of three out of the eight species. When comparing these ficticious scenarios, it becomes evident that all predefined land-sharing systems would yield the same timber yield as a land-sparing system consisting of a 50% of monospecific stand and 50% conservation area. Given this specific condition of equality, we could easily deduce the superior choice between the sparing and sharing systems based now on their performance regarding total carbon storage and/or Shannon biodiversity levels.

Now if we are considering above-ground living carbon biomass as the criterion for comparison, it is essential to factor in the carbon content of monospecific carbon planting and the spared forest within the context of land sparing. Subsequently, the total of these carbon contributions should be compared with the carbon stock on the shared land.

It is worth noting that for the projections presented in this article, the model exclusively calculated local ($\alpha$-)biodiversity, which remained constant regardless of the size of the conservation area. Nonetheless, one could argue that in certain contexts, allocating 50% of the land to a conservation area might be less appealing than achieving the same level of $\alpha$-biodiversity with a land-sharing system occupying the entire land area. Conversely, it might be considered more attractive if the $\alpha$-biodiversity achieved with a land-sharing system is lower. However, addressing this issue is beyond the scope of this article.

### Hypothesis regarding the expected outcome

After a comprehensive comparison of all conditions, it was anticipated that the ratio of spared land would theoretically fluctuate based on both the monospecific stand and the 256 land-sharing strategies applied. This variability was expected to mirror distinctions in life-history traits among species, as these traits are recognized to influence demographic performance and coexistence outcomes (*Pichancourt et al., 2014*; *Pichancourt, 2023*). Nevertheless, due to the intricate nature of this relationship, it was considered challenging to make any assumptions regarding the establishment of a straightforward ranking between sparing and sharing.

### Model code to reproduce the results or study other systems

The model was written using Gnu-Octave (*Eaton et al., 2023*), a free and open source software fully compatible with MATLAB (*The Mathworks, Inc., 2022*). The code developed in *Pichancourt et al. (2014)* and *Pichancourt (2023)* and used to produce the present simulation results is freely available *via* the original url provided at the end of this article. A comprehensive presentation of simulation results can also be found in File S1.

## RESULTS

### Land-sharing performances

Under these methodological conditions, the forest land-sharing model demonstrates for the level of biodiversity, that not granting access (*i.e.*, creating a conservation area) is predicted to lead to the best result in term of biodiversity. This result would be equivalent to the conservation area in the land sparing system ($RSI_C$). However, as soon as access is granted, having greater proportion of tree species managed by loggers (from 12.5%

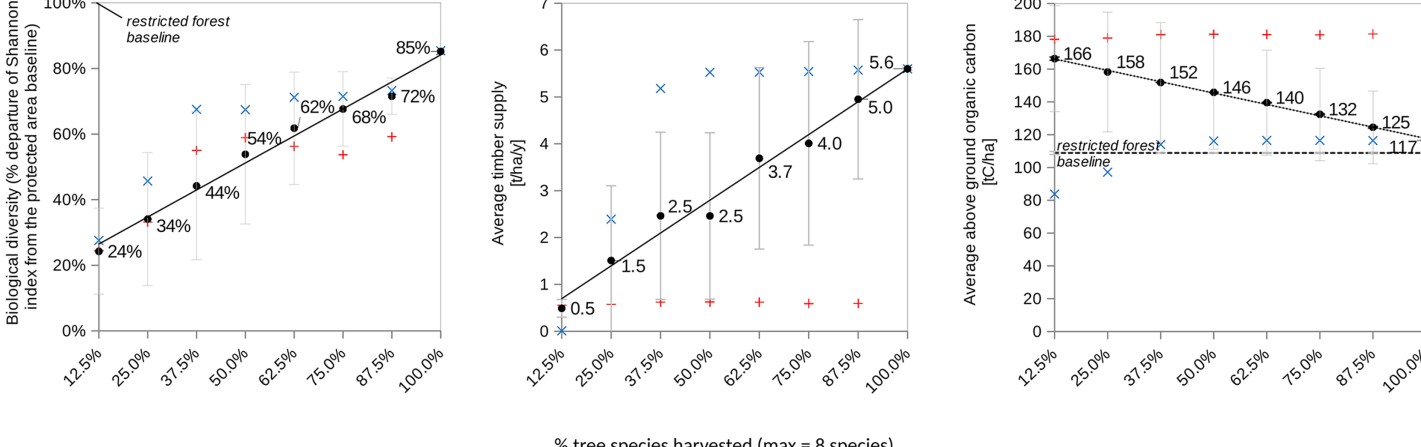

**Figure 2 Illustrates the performance trends over a 100-year simulation for various forest land-sharing systems (refer to Fig. 1A and (insert appropriate reference)) characterized by distinct levels of access granted to independent exploiting actors EAI.** These actors are engaged in logging and replanting each one of the eight tree species present in the forest. A total of 100% species harvested means eight species logged and replanted by eight actors. The figure utilizes blue and red labels to depict sensitivity analyses related to regulating access to the shared forest (variable 5a). Specifically, it explores two different constraints on logging (1b') and replanting (1a') for the tree species that exhibited the highest productivity under the land-sharing arrangement (*i.e.*, sp#4 from Table 1). The red label signifies the scenario of not logging sp#4, while the blue label represents the systematic logging of sp#4. Error bars in the figure denote standard deviation.

(one species) to 100% (eight species)) is expected to yield collectively non-additive synergistic effects among pairs, triplets, and up to octuplets of dyads as described in Table 1. Consequently, socially more diverse shared forests are predicted to simultaneously enhance tree Shannon biodiversity, above-ground carbon storage, and collective timber supply (Fig. 2).

A sensitivity analysis was performed (Fig. 2) that consisted in assuming SAI forbids (red dots) or mandates (blue dots) EAI to harvest the most productive species sp#4. The results show that systematically logging and replanting the most productive and dominant tree species significantly increased both timber yield and the Shannon index of tree diversity, but significantly reduced the above-ground living carbon biomass. Conversely, forbidding its harvest had the opposite effects on all the forest variables. This confirmed that the dominant species plays a significant role in driving the levels of biodiversity, carbon storage, and timber Yield.

## Comparison with land-sparing

The outcomes of various sharing scenarios were compared with land-sharing scenarios containing one of the eight monospecific stands (Tables 2A and 2B). The results indicate that the choice between sparing or sharing depends on factors such as the selected species for the monospecific stand, the chosen objective for the stand (carbon storage, timber yield), and the number of species harvested in the shared forest.

Comparing the outcomes in Tables 2A and 2B to the insights from Fig. 2, it is evident that a one-size-fits-all solution for land sparing or sharing is elusive. Nevertheless, practical guidelines can be derived to inform the choice between these approaches.

**Table 2 Percentage of land that can be spared for conservation (biodiversity) when comparing to various land-sharing systems based on different criteria for comparison: (a) the total forest above-ground carbon stock is used and (b) the average annual timber harvested, estimated.** Within these two options of comparison, percentages can change with the type of mono-specific plantation (using species sp#1 to sp#8 presented in table rows), the type of constraint imposed on harvesting in the land-sharing system (columns), the number of species harvested (from 0 to 8 for the red sparklines reading from left to right) in the land-sharing system (sparklined histograms with eight bars max for every table cell). The values were estimated by comparing the average above-ground forest level carbon stock over 100 years of simulations for every mono-specific stand Table 1 and for land-shared forests described in Fig. 1. The exact percentage of spared land from histograms are presented in Table S1 and Table S2 from Appendix S2. The tallest histogram bars reach 100% of land-sparing potential (*i.e.*, allowing 100% of land spared for conservation): *e.g.*, land-sparing with sp#3 *vs* land-sharing with "sp#4 always" with only one logger with access granted (left side of the sparkline) under criteria of comparison (b). When there is no histogram bar, it means that no land (0%) can be spared for biodiversity. Simulation results and analysis to produce the table can be found in the File S1.

| Land-sparing with Mono-specific stand* | Land-sharing with 0 to 8 species harvested and different constraints | | | | |
|---|---|---|---|---|---|
| | no constraint | sp#4 always | sp#3 always | sp#4 never | sp#3 never |
| | (a) Criteria for comparison: above ground carbon stock [$tC.ha^{-1}$] | | | | |

| Land-sparing with Mono-specific stand* | no constraint | sp#4 always | sp#3 always | sp#4 never | sp#3 never |
|---|---|---|---|---|---|
| (b) Criteria for comparison: average annual timber harvested [$t.ha^{-1}.y^{-1}$] | | | | | |

It is noteworthy that planting monospecific stands to maximize above-ground living carbon stock (Table 2A) is expected to yield less favorable land-sparing results compared to planting for timber yield (Table 2B).

When carbon storage is used as the comparison criterion, most monospecific stands fail to match the performance of different land-sharing systems (Table 2A), leaving no

available land for conservation purposes in such cases. For instance, all land-sparing systems with a monospecific stand of species sp#1, 2, 4, 6, 7, or 8 could never produce as much carbon as any land-sharing scenario (Table 2A). Conversely, for monospecific stands outperforming land-sharing systems, sparing land for conservation becomes feasible, as indicated by non-zero percentages in Table 2A. This is observed only for sp#3 monospecific stands planted for carbon.

Furthermore, land-sparing containing monospecific stands of sp#3 planted this time for timber would perform even better compared to many land-sharing scenarios. Monospecific timber planting with sp#1 and/or sp#4 would also be of interest, but to a lesser extent, for sparing land (Table 2B), but only when compared to land-sharing systems where a very low proportion of species are managed independently by loggers (left side histogram bars in sparkline).

Under conditions where timber yield is the criterion for comparison and the monospecific stand consists of the high-performing species sp#3, land-sparing appears to be a more straightforward choice. This effect is more pronounced when compared to a land-sharing system where the most performing species sp#4 is never harvested (Table 2B). Conversely, land sharing should involve a greater proportion of species managed by a more diverse group of sustainably cautious loggers (refer to Table 2B and Fig. 2) to ensure land-sparing is rarely an attractive option.

The sensitivity analysis, involving the prohibition or mandatory inclusion of sp#4 in harvesting, corroborated results on land-sharing. The preference for land-sharing over land-sparing becomes more pronounced when sp#4 is consistently harvested. Prohibiting its harvest in a land-sharing scenario would result in a suboptimal sharing system, reaching a point where an equivalent performance could be achieved by adopting a land-sparing system with less than 10% of a sp#3 monospecific stand and allocating over 90% of the land for conservation, including the habitat of sp#4.

Given the crucial role of sp#3, additional sensitivity analyses were conducted by examining land-sharing systems where the harvest of sp#3 was either prohibited or mandated. The results did not significantly affect land-sparing systems featuring sp#3 monospecific stands, but noticeable changes were observed when contrasting land-sparing systems featuring sp#1 or sp#6 monospecific stands with land-sharing systems where only a limited number of loggers were granted access to the shared space (Table 2B).

## DISCUSSION AND CONCLUSION

### Summary of results and general intepretation

This article presented a social-ecological system modelling framework to be able to (i) model the fundamental difference of social-ecological structure between land-sharing and land-sparing, (ii) predict forest biodiversity and ecosystem services outcomes for different social-ecological contexts, and (iii) rank a great variety of land-sharing and sparing solutions, based on their respective capacity to optimize the level of different ecosystem services and of biological diversity.

A first outcome of this study underscored the highly context-specific nature of the sharing-sparing problem, making it challenging to definitively favor one approach over the

other. Both approaches held theoretical merit, with the selection of the most suitable solution contingent upon several factors, including the species composition and functional traits, specific objectives and quantitative criteria, and the prevailing social structures and governance rules that regulate forest access and various individual and collective management actions (see Fig. 1).

Despite the intricacies of the social-ecological dynamics considered, the results highlighted the emergence of simple heuristics that could aid in optimizing both land-sparing and land-sharing preferences. For instance, it became apparent that a monoculture of a suboptimal species is unlikely to perform as well as a land-sharing system, particularly when the latter achieves perfect coexistence among a diverse array of species and appropriators. However, the model also lends support to critics of land-sharing who argue in favor of sparing as a safer option (*Fischer et al., 2014*; *Runting et al., 2019*). These critics contend that, even in scenarios with fully cooperative loggers, the presence or absence of collective constraining rules by the SAI (Supporting Actors & Infrastructure) related to threatened or dominant species can significantly impact biodiversity, carbon levels, and timber outcomes, as confirmed in Table 2 (see sensitivity analysis with constraints imposed on species logging). This sensitivity analysis reinforces the notion that the success of land-sharing hinges on the existence of appropriate collective choices and knowledge within a community within SAI.

In our model, these the constraining rules were particularly associated with the regulating processes 5a, 5a', 5b, and 5b' presented in Fig. 1 and the Eqs. (3) and (4) of the simplified macro-model. The role of these SAI processes aligns with previous assumptions made by various authors (*Parrotta & Agnoletti, 2012*; *Agrawal, Wollenberg & Persha, 2014*; *Jiren et al., 2018*), underscoring the fact that if these regulating processes cannot be implemented, then land-sharing may appear to be a riskier choice compared to the simpler and safer land-sparing approach. These findings therefore imply that selecting the most suitable approach cannot solely rely on performance metrics, reinforcing the idea that cultural preferences and social feasibility should also be taken into account (*Fischer et al., 2014*; *Mastrangelo & Laterra, 2015*). The distinction lies in the demonstration that we can now systematically assess the influence of these cultural preferences when comparing various options for sharing and sparing.

## Consequences for the analytical theory of sharing-sparing

Previous approaches employing the optimization paradigm were rooted in the concept of land-use intensity, through bioresource exploitation such as logging for timber in our case (*Green et al., 2005*; *Butsic et al., 2012*; *Fischer et al., 2014*; *Phalan, 2018*; *Runting et al., 2019*). These models assumed a negative impact of harvesting on local biomass as well as biodiversity per unit of time, implying inevitably a tradeoff relationship between timber yield and living biomass or biodiversity levels. Building on this assumption, previous analytical and numerical findings suggested that solutions closer to land-sparing should take precedence to optimize this trade-off. However, the validity of this assumption was questionable (*Pichancourt et al., 2014*; *Pichancourt, 2023*), and when reevaluated like it was done here with this new class of SES model, it was found to lead to counterintuitive results

concerning the relationship between forest outcomes, consequently impacting the rankings between sharing and sparing. These have two consequences for existing analytical sparing-sharing theory.

The first consequence is related with the asumption made on the *minus* sign to describe the negative impact of land-use intensification on local ecological outcomes, cf. Eqs. (1) and (3). In our case we intensionaly used the ± sign when presenting the simple analytical model. When using the detailed mechanisms in the micro-model, the results confirm mostly the negative impact on above-ground carbon biomass (*minus*, see Fig. 1). However, they not confirm it on the Shannon level of tree diversity. For the latter, we did not observe an intermediate level of disturbance as suggested in (*Butsic et al., 2012*). Although this intermediate disturbance hypothesis could in theory emerge in other contexts with the detail model used here, we rather non trivially predict a positive correlation between harvesting intensification and local level of biological diversity. Nevertheless, important caveats need to be reminded here to avoid any confusion that would lead to the belief that openning the gates to as many loggers in forests is a good idea, which obviously so many facts from the literature prove that it is not. Intensification in our case was described as an emmerging process of selective logging by decentralized actors that were each specialized on sustainably logging and replanting a single species each (cf. method section). So more intense means greater proportion of species sustainably, yet independently, harvested. This process is more in line with some of the social organizations observed in many land-sharing systems (*Natcher, Davis & Hickey, 2005*; *Lemos & Agrawal, 2006*). Consequently, future analytical studies on sharing-sparing will have to consider a range of parametric relationship between land-use intensity and forest outcome, greater than the one used *e.g.*, in *Butsic et al. (2012).*

The second consequence is related to the control of processes associated with the Supporting Actors and Infrastructure (SAI). This study examined specific controls pertaining to access rates (5a, 5b) for logging (5a') and replanting (5b') (see Fig. 1 and Eqs. (1)–(4)). These controls were used to impose constraints on the access, logging, and replanting of the most productive species sp#4. These constraints fundamentally altered the relationship among timber yield, carbon storage, and biodiversity, consequently influencing the sparing-sharing ranking. Interestingly, consistently logging the dominant species favored land-sharing preferences, whereas refraining from harvesting it resulted in the opposite preference for land-sparing. These findings, while specific to the current study and reliant on a detailed micro-model, underscore the same necessity of controlling the range of parametric relationship between land-use intensity and forest outcome (1b in Eqs. (1)–(4)). However, it also amphasizes the fact that this control must be precisely explained in terms of on-ground logging practices and their relation with controlling processes. Currently as it stands, the generic term "land-use rate" used in the sharing-sparing literature, lacks these meaningful implications and prove here to not be of great value for understanding many situations associated with selecting between sharing and sparing. However, if through the exploration of a great range of case scenarios involving precise land-use/logging practices, it becomes possible to emerge general functional rules regarding the parametric relationship between forest outcome, land-use intensity (1b) and

its control (5a, 5a', 5b, 5b'), then there is potential to simply use existing analytical study to generate more comprehensive and useful ranking solutions.

## Limits and extrapolation of results to more complex systems and other taxa

The rankings for land-sparing and land-sharing were determined using a model with specific assumptions detailed in the methodology section. Some of these assumptions made in the model are amenable to relaxation, such as perfect monitoring or support, and absence of water limitation. The logic presented before could be extended to also include all the other linking variables presented in Fig. 1 and the one associated with the fourth unused compartment mentionned in the method section, then the future analytical theory of sparing-sharing would align and benefit more closely from the powerful the governance theory of SES as articulated by E. Ostrom and Anderies (*Ostrom, 1990*; *Anderies, Janssen & Ostrom, 2004*; *Anderies, Barreteau & Brady, 2019*). The logical implication from what has been presented here is that the sets of Eqs. (1) and (2) could be used as a new reference frame for such general theory.

Other processes haven't been studied here, including the sensitivity to climate scenarios, change in soil depth and structure, and the number of species in the forest. These aspects can directly be modeified in the model and code (see link presented in method section). Additionally, new factors could be introduced in the model and code that would help compare the results with previous studies. For instance, the present model only deal with the sparing-sharing ranking based on tree species biomass and diversity. However other studies have also tried to compare the impact on the diversity of other taxa (*Marull et al., 2015*, *2018*; *Runting et al., 2019*). A simple way to achieve this in the model would be to model the statistical relation between tree or forest variables and the biomass or diversity of other taxa. For instance, we could simply model the correlation between beak gape and seed/fruit size of the tree species (see Table 1) to generate an indicator of bird diversity (*Wheelwright, 1985*). This way we could more easily compare the present results with previous ones (*Marull et al., 2015*, *2018*; *Runting et al., 2019*).

While all these parameters and variables would likely play a role in the land-sparing and sharing rankings, certain factors may exert more influence than others. To gain a comprehensive understanding, a sensitivity analysis is necessary. This would elucidate whether general principles persist when the problem is made more intricate, or if the land-sparing and sharing debate is inherently context-specific. In the latter case, the objective would be to formulate simple, case-by-case predictions and comparisons. People could then define their cultural preferences accordingly (see *e.g.*, *Mastrangelo & Laterra, 2015*).

## Toward a more fundamental theory of sharing *vs* sparing

The present article demonstrated that the debate between sparing and sharing is highly dependent on how we frame the land-use intensity problem, especially in relation with land-sharing system. Given the complexity of these processes, it was offered in the previous sections some suggestions for simplification in order to include other interesting variables

in relation with governance theory. However, it is not trivial to expect a-priori to expect that general rules would emerge too. This challenge has been extensively discussed in past studies to justify the need to utilize simpler analytical trade-off approaches (*Phalan, 2018*). Navigating between these opposing needs poses a significant question. One approach, as we will suggest, would involve examining how land-sharing and our non-intuitive findings can be framed within control theory itself.

In fact before delving into the results, one might easily have presumed that a land-sharing system, fostering increased access for a more diverse array of loggers, would unlikely outperform numerous land-sparing alternatives. If we reverse the argument, why should we anticipate that the coexistence of a greater social diversity of independent yet sustainably cautious species loggers/controllers would enhance biodiversity, carbon sequestration, and timber production, to the extent that it surpasses most land-sparing options? At present, invoking only the widely used optimization paradigm from control theory would be difficult (*Green et al., 2005*; *Butsic et al., 2012*; *Fischer et al., 2014*; *Phalan, 2018*; *Runting et al., 2019*).

This result can in fact be elucidated by invoking Ashby's much simpler "Law of requisite variety", also from control theory (*Ashby, 1958*; *Porter, 1976*). This well-known phenomenological law posits that, for effective control and stabilization of a system, the controlling entity (in this case, the EAI with its diverse independent appropriators acting through 1a, 1a', 1b' and 1b') and the stabilizing entity (in this case, the SAI comprising the same actors but acting for the benefit of the community and the RSI *via* the other linking processes) must possess a level of "variety" (*i.e.*, diversity) at least equal to that of the system being controlled (*i.e.*, the number of species in the RSI). Below this threshold, Ashby's law would predict suboptimal control on forest outcomes. The present results for land sharing serves as a concrete illustration of this law's prediction. Therefore, while Ashby's law is phenomenological and does not establish causality, when associated with the outcomes of the present mechanistic model, it provides indication about potential social-ecological mechanistic pathways for understanding why this law may be universally effective to solve the sparing-sharing debate.

This interpretation would not have been possible without the pioneering work of *Ostrom (1995)*, who unexpectedly was the only one to hypothesize the role of Ashby's law to explain the frequently observed positive impact on biodiversity outcomes of increasing polycentricity in governance (*i.e.*, a greater diversity of distributed controllers as per *Ostrom, 1990*). The present study not only resurfaces and corroborates Ostrom's forgotten governance hypothesis on SES, but also indicates a deeper connection to a more fundamental debate within the field of control theory applied to SES. Specifically, it relates to the choice between Ashby's Law of Requisite Variety (for local land-sharing and land-sparing diversity at a landscape scale) and optimal control methods (*i.e.*, optimal monospecific stands and harvesting for land-sparing) (*Porter, 1976*).

To solve this new debate, it would be necessary to conduct a proper analytical comparison between them across a broader spectrum of species (including using other taxa), soil compositions, and climatic circumstances. For instance, it would be valuable to compare the respective benefits of Ashby's law predictions with the outcomes of the

present model assuming changes impacting different control channels, *i.e.*, the links 1a to 6b between compartments depicted in Fig. 1 and associated equations. More specifically, links 0 to 1 within and between RIS and EAI should directly influence the interplay between the diversity of direct controllers (appropriators) and controllees (species), whereas links 5 to 6 should have an impact on the stability of the relationship. A promising idea would be to compare these two control approaches within the framework of landscape-scale sparing-sharing theory (*Marull et al., 2015*, *2018*), or within the framework spatial planning, which is also still very much influenced by the optimal control paradigm (see *e.g.*, *Venter et al., 2013*). For the latters, Ashby law may perform better or reveal non-trivial sharing-sparing options, especially when more polycentric arragements–involving different public, private and community actors–are designed to control spatially a greater diversity of species involved in the provisionning of many ecosystems services (in our case through 1b').

In any event, conducting a meticulous exploration of this issue across a wider range of social-ecological contexts could offer a promising avenue to develop a more comprehensive theory of land-sharing/sparing. Such a theory might prove more beneficial for decision-making processes.

## CONCLUSION

This article introduced a SES modeling framework of the sharing *vs* sparing nexus rooted in the CISF theory. The primary objective of the framework was to capture essential analytical distinctions in social-ecological structures and dynamics between land-sharing and land-sparing. It then seeked to forecast outcomes for forest tree biodiversity, carbon storage, and timber yield across diverse social-ecological contexts, ultimately ranking various land-sharing and sparing solutions based on their effectiveness in optimizing these forest outcomes on the long-term.

To ensure a meaningful comparison with prior analytical studies, the model deliberately confined its scenarios and results to a narrower set of variables, with a specific focus on certain CISF governance constraints and forest biophysical conditions. This approach facilitated an in-depth exploration of the identified factors and shed light on their influence on forest outcomes.

The study revealed specific deficiencies in prevailing land-sparing and land-sharing theories. It emphasized the importance of incorporating existing social and ecological theories that delved into the intricate connections between disturbance practices, productivity, and ecological outcomes, unveiling nuanced processes that significantly impact the performance of land-sharing and land-sparing approaches.

By addressing these shortcomings, the study not only provides valuable insights but also establishes the groundwork for a more comprehensive analytical framework to compare land-sharing and land-sparing performances across a wider spectrum of governance systems, biophysical conditions and taxonomic groups not explored in this study.

## ACKNOWLEDGEMENTS

I express my gratitude to Pr. J. Marty Anderies for engaging in fruitful discussions regarding the minor adjustments of the CISF, ensuring its alignment with the present objectives and case study.

### Funding

Jean-Baptiste Pichancourt was supported for this work by the Institut National de Recherche pour l'Agriculture, l'Alimentation et l'Environnement (INRAE). The funders had no role in study design, data collection and analysis, decision to publish, or preparation of the manuscript.

### Grant Disclosures

The following grant information was disclosed by the authors:
Institut National de Recherche pour l'Agriculture, l'Alimentation et l'Environnement (INRAE).

### Competing Interests

The author declares that they have no competing interests.

### Author Contributions

- Jean-Baptiste Pichancourt conceived and designed the experiments, performed the experiments, analyzed the data, prepared figures and/or tables, authored or reviewed drafts of the article, and approved the final draft.

### Data Availability

The raw results of model simulations are available in the Supplemental File. The original model used to produce these results is available at: https://doi.org/10.7717/peerj.14731/supp-1.

The mathematics of the model of multi-species forest ecosystem dynamics (RSI) is available in the Supporting Information (Data S1 and Tables S1 and S2): Pichancourt, J.-B., Firn, J., Chadès, I. and Martin, T.G. (2014), Growing biodiverse carbon-rich forests. Glob Change Biol, 20: 382-393. https://doi.org/10.1111/gcb.12345.

The mathematics of the exploiting (EAI) and supporting actors and infrastructures are available in the Methods: Pichancourt J. 2023. Some fundamental elements for studying social-ecological co-existence in forest common pool resources. PeerJ 11:e14731. https://doi.org/10.7717/peerj.14731.

### Supplemental Information

Supplemental information for this article can be found online at http://dx.doi.org/10.7717/peerj.16809#supplemental-information.

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
