# Peer review of "Navigating the complexities of the forest land sharing vs sparing logging dilemma: analytical insights through the governance theory of social-ecological systems dynamics"

_PeerJ, doi:10.7717/peerj.16809_

## Round 0.1 · original submission · Major Revisions

Please do a major revision according to the suggestions of the two reviewers. A clear point-by-point response is needed. The following points should be specifically addressed.
- Add a clear justification on uniqueness that warrants publication as a separate paper. particularly, how is this adding value compared to the previous publications. This is unclear.
- Add more clarity to your research presentation.
- Details on methods should be clear with less jargon

Reviewer 1 ·

Basic reporting

English language is neat, well written, clear, showing professional English. The structure of the manuscript, if it is a Review paper, is confusing and does not follow PeerJ instructions. There are no subtitles.
Taking a look in PeerJ web page in instructions to the authors, for a review paper the authors must include an Abstract, I found the abstract for this paper, but not an introduction neither a conclusion. Therefore an orientation for the reader is missing. There is no information about raw data.
The figures are relevant, clear and good quality.

Experimental design

There is no definition of the methodology, the author suggest the reader to read his papers for understanding the resolution of the models.

The methods are not described, that is why I assume it is a Review paper.

The author does not indicates where the data is taken for building the models, is all “invented data” or is taken from a data base or a platform, the author should mention that important point.

Validity of the findings

Scientific relevance of the Manuscript
More specific information is needed in the manuscript, reviewers are forced to find relevant details in Pichancourt (2023),and Pichancourt et. al (2014) in order to understand the discussion and findings of the paper. The manuscript is permanently referring the reader to his previous papers Pichancourt (2023),and Pichancourt et. al (2014). The reader is obliged to read those papers for understanding the methodology used for the models and for the analysis as well.
The manuscript contributes to knowledge in the sparring versus sharing methods of conservation. The newly sparing-sharing strategies could be most effective for the conservation of forest systems.

Additional comments

If it is a review paper more references are needed

Reviewer 2 ·

Basic reporting

The study in general needs more detail and context. References to your previous study are frequent but the study should be able to be interpreted as it is. I suggest that you improve the description in the beginning and provide more justification for your study. Importantly, adding a general objective/Research question of the study would be useful to guide the readers, since as it is it gets hard to follow. The same suggestion of more context goes for the tables and figures. They contain very specific, detailed information that is difficult to interpret without more information in the main manuscript. The text should finish with a strong conclusion of what is the relevance and what are the contributions of this specific study to the sparing-sharing issue.

This manuscrit has been submitted as a research article, yet it does not include the expected sections: introduction, materials and methods, results and discussion (see comment on basic reporting).

Experimental design

I think the study addresses a very relevant issue (whether , exploratory models can produce simple prior evaluations and heuristics to define the right strategy across a broader spectrum of social-ecological contexts). But it is not easy to read and straightforward with its current structure, that to begin with does not meet the standards of the journal.

Validity of the findings

It is difficult to evaluate ethe validity of the findings with missing information on how these results were generated. I therefore strongly suggest incorporating all the needed sections to the manuscript.

Additional comments

General minor comments on text:
L. 17: As per PeerJ author guidelines references are not encouraged. I suggest removing it.

Table 1. I think it should be moved a little below (e.g. after L.60).ç
L.27-36. When defining land-sparing and sharing I think how land-sparing is framed is a bit too specific. It could be reframed into something like “In contrast, land-sparing emphasizes spatial segregation at the landscape scale of production and other uses (e.g. conservation, recreation) (e.g. segregation of several optimally managed mono-specific stands, each specializing in different livelihood benefits).

L.51-53: I suggest to add more context regarding the social-ecological forest model and results produced by Pichancourt (2023, what does the model consist in, what kind of results are produced? Otherwise it gets difficult to understand how it is transferrable to other situations. The same goes for the statement “Next, all numerical results used for the sharing-sparing inferences are presented in Appendix S1”.

Obs: I did not make specific comments after L.80 since I consider it is mandatory to properly structure the article in its different sections.

---

## Round 0.2 · Major Revisions

The manuscript still needs a thorough revision for clarity.

- Model needs better explanation in the context of approach, explanation of equations used, comparison to other models, and what is unique compared to existing models

- introductions need to be recast in the context of related studies by including relevant citations

- Objectives and hypotheses need to be clear and how they are tested and validated

-Methods need to be clearly set in the context of literature

- Shortcomings of the model and assumptions used in the approach need to be discussed clearly and acknowledged in light of potential misinterpretation

- discussions need to situate in the context of other related studies, which is insufficient as written.

- It is still not clear if this version is a substantial contribution to the earlier paper of the author. If this is not justifiable, it will not raise to the standard for publication as a new contribution.

Reviewer 2 ·

Basic reporting

The paper has been improved in relation to the previous version, yet some key elements are missing for a proper understanding of the research questions, methods and results. Contributing to the sparing/sharing debate with a specific SE model that allows comparing multiple numerical outcomes under different scenarios is valuable and novel. Yet as it stands the article is not easy to understand and follow.

In the introduction, more background is needed particularly on specific studies that have addressed land sparing/sharing for different taxonomic groups. In the methods, more clarity and explanations about the model and ho is it built are highly needed. A clear definition of the objective(s) is also highly encouraged.

In the Discussion, putting the results in the conext of other studies evaluating other taxonomic groups and contexts would be desirable, beyond going through the specific results of this study.

Experimental design

Comments made in attached document.

Validity of the findings

Due to the lack of clarity on key aspects (e.g. model building) in the Methods section it is hard to evaluate the validity of the findings.

Annotated reviews are not available for download in order to protect the identity of reviewers who chose to remain anonymous.

---

## Round 0.3 · Minor Revisions

ract- Revise abstract to simplify reference to Ostrom - Say, "The findings also support a governance hypothesis of Ostrom (2009) regarding the positive impact on biodiversity and productivity of increasing polycentricity.
- Include a paragraph in the introduction that outlines the differences between your 2023 paper and this paper, as stated in the rebuttal.
- The conclusion section is missing. Even though it is included in the earlier section, the conclusion section should be separate and clearly summarize key findings and shortcomings. In this section, highlight the shortcomings of this paper and add suggestions for future research.

---

## Round 0.4 · accepted · Accept

I am recommending publishing the article.